# Consumption of alcohol and binge drinking among pregnant women in Addis Ababa, Ethiopia: Prevalence and determinant factors

Mezinew Sintayehu Bitew[1]*, Maereg Fekade Zewde[2], Muluken Wubetu[3], Addisu Alehegn Alemu[4]

1 Department of Mental Health, Debre Markos University, Debre Markos, Amhara, Ethiopia, 2 Department of Maternal and Child Health, Addis Ababa City Administration Health Bureau, Addis Ababa, Ethiopia, 3 Department of Pharmacy, Debre Markos University, Debre Markos, Amhara, Ethiopia, 4 Department of Midwifery, Debre Markos University, Debre Markos, Amhara, Ethiopia

* sintayehume@gmail.com

## Abstract

### Introduction

People in Ethiopia, including pregnant women, highly consume both home-made and manufactured alcohol beverages due to lack of awareness about the harmful effect of risky alcohol use, and cultural acceptance of alcohol consumption. Alcohol consumption and other hazardous patterns of use like binge drinking have tremendous adverse effects on fetus and mothers. Therefore, this study aimed to assess the magnitude of alcohol consumption, binge drinking and its determinants among pregnant women residing in Kolfe sub-city, Addis Ababa, Ethiopia.

### Methods

Institutional based cross-sectional study was conducted among a total of 367 pregnant women. The participants were selected using a systematic random sampling method. Data were collected through a structured questionnaire. A binary logistic regression was conducted using SPSS version 20 software to identify determinants of alcohol consumption and binge drinking. A p-value < 0.05 was used to declare a statistical significance in multiple logistic regression. The results were described using adjusted odds ratio with a 95% confidence interval.

### Results

This study revealed that the prevalence of alcohol consumption, binge drinking, and weekly alcohol consumption of four or more units among pregnant women was 39.78%, 3.54% and 4.9%, respectively. Not having formal education [AOR 95% CI = 8.47 (2.42, 29.62), having primary education [AOR 95% CI = 4.26 (1.23, 14.74), being a housewife [AOR 95% CI = 4.18 (2.13, 8.22), having an unplanned pregnancy [AOR 95% CI = 2.47(1.33, 4.60), having a history of abortion [AOR 95% CI = 3.33 (1.33, 6.05)], not having awareness about the harmful effect of alcohol consumption [AOR 95% CI = 4.66 (2.53, 8.61)], and not having

**Data Availability Statement:** All relevant data are within the manuscript and its Supporting Information files.

**Funding:** The authors received no specific funding for this work.

**Competing interests:** The authors have declared that no competing interests exist.

**Abbreviations:** ANC, Ante Natal Care; AOD, Alcohol and Other Drugs; AOR, Adjusted Odds Ratio; COR, Crude Odds Ratio; CDC, Center for Disease Control; CI, Confidence Interval; EDHS, Ethiopian; Demographic and Health Survey; ETB, Ethiopian Birr; FASS, Fetal Alcohol Spectrum Syndrome; HC, Health Center; MSPSSf, Multidimensional Scale of Perceived family Social Support; WHO, World Health Organization.

family social support [AOR 95% CI = 2(1.14,3.53) were determinants of alcohol consumption among pregnant women.

## Conclusions

This study found a high level of alcohol consumption among pregnant women. Interventions to create awareness on the harmful effects of alcohol are needed. Moreover, strengthening social support during pregnancy and family planning services to reduce unplanned pregnancy and abortion should be considered.

## Introduction

Globally, use of substances such as alcohol, tobacco and other illicit substances, in a way that could harm the users' health, have escalated through time in all segments of the population and has become one of the growing public health and socioeconomic concerns. Alcohol shares the highest-burden that accounts for 3.8% of all global deaths and 4.6% of the global burden of diseases and neuropsychiatric disorders [1–3]. Risky use of alcohol causes significant morbidity and mortality (particularly from injuries) and societal harm such as social disruption from crime, unemployment and marital disharmony [2, 3].

The negative health consequences of alcohol consumption are more profound for pregnant women and their fetuses. Alcohol consumption during pregnancy has also shown to be one of the leading causes of preventable birth defects and developmental problems comprising language and motor delays and deprived academic achievement [4–6].

New studies indicate that even low levels of prenatal alcohol exposure could adversely affect the developing fetus. Cognitive and socio-emotional deficits, among children exposed to even small amounts of alcohol, are found [7]. Consumption of 2 or more standard drink of alcohol/day in early pregnancy is associated with significant problems on the fetus. It can cause miscarriage, preterm birth and low birth weight compared to children born from non-drinkers. In addition, other studies reported that women who consumed more than three drinks per week during the first trimester had a significant harmful health effect on their pregnancy. The consequence would be five folds more if they took five or more units of alcohol per week [8–12]. Furthermore, binge-drinking (commonly defined as consuming 4 or more units of alcohol at one occasion for women) during pregnancy is more harmful because of its serious toxic effects on fetal neurodevelopment crossing the placenta [13, 14].

The adverse effects of prenatal alcohol exposure can range from subtle developmental problems, or fetal alcohol effects, to full-blown fetal alcohol spectrum syndrome (FASS). However, it is not limited to infancy and childhood; prenatal exposure to alcohol, particularly in early pregnancy, has also been found to increase the likelihood of developing an alcohol disorder in adulthood [15, 16]. Given these consequences and the belief that there is no safe time or amount to consume alcohol in pregnancy, the CDC recommends complete abstinence [17, 18].

Studies have reported alcohol consumption among pregnant women ranging from 19.5% in South Africa to 59% in Nigeria [19–26].

Similarly, studies conducted on the prevalence of binge drinking among pregnant women showed between 3% in Canada and 25% in Congo [3, 27–31]. A study in New Zealand also revealed 10% of pregnant women drank seven and more units of alcohol per week which is the other clinically significant patterns of alcohol use [30].

According to studies from different countries showed that socio-demographic factors (age, marital status, educational status, and occupation), obstetric history (gestational age, unplanned pregnancy, gravidity, and history of abortion), social support and behavioral issues like use of Khat and cigarette smoking were determinants of alcohol consumption during pregnancy [32–37]. Additionally, studies reported that awareness about the harmful effect of alcohol on fetus, attitude and culture of the population were determinants of alcohol consumption [36, 38–40].

Alcohol consumption and its associated problems among pregnant women increased globally [41]. In low-income countries including Ethiopia, the rapid increase of the consumption of alcohol has also been indicated for decades, especially reproductive-age women are affected [1, 42]. Both traditional and commercially produced alcohol beverages are widely spread more than ever which are also an important contributing factors for increased alcohol consumption in Ethiopia [1, 38].

People in Ethiopia that include pregnant women highly consume alcohol beverages, especially the traditionally prepared alcohol beverages like Areki, Teje, Tella that have higher percentages of alcohol amount [43–45]. Thus, locally made alcohols are being used excessively without control in different holidays, religious festivals, parties, and almost in daily family meals and day to day life [43–45].

Despite the adverse health effects of alcohol consumption is profound, its assessment with determinants among pregnant women is limited. In particular, there is a scarcity of information on binge drinking of alcohol by measuring or calculating the amount consumed. Addis Ababa, the capital city of Ethiopia where the current study was conducted, is heavily affected with consumption of alcohol [1]. Therefore, this study was intended to assess the magnitude of alcohol consumption and binge drinking among pregnant women, and its determinants in Kolfe sub city, Addis Ababa, Ethiopia. The finding of this study will have significant contributions for health program managers, policymakers, clinicians and researchers.

## Materials and methods

### Study design, sample size and sampling technique

An institutional-based quantitative cross-sectional study was conducted based on the guidelines of strengthening the Report of Observational Studies in Epidemiology (STROBE) for observational research [46]. It was conducted at Kolfe Keranio sub-city, Addis Ababa, Ethiopia. The sub-city covers an area of 61.25 sq.km and has a total population of 546, 219 (235, 360 females). Around 2.33% (9,531) of females were pregnant. The sub-city has 15 districts and 11 Health Centers (HCs) [47].

A total of 397 pregnant women were included in this study. Single population proportion formula was considered to calculate the sample size. Taking 34% prevalence (p) of alcohol consumption among pregnant women [48], 5% margin of error (d), 95% confidence interval (CI) and 15% non-response rate. Systematic random sampling was carried out after the sample was proportionally allocated to each of the 11 health centers. All pregnant women, regardless of the gestational age, attended the health centers during the study period were included. However, pregnant women who were seriously ill and unable to respond were excluded. The study was conducted from May to June 2017.

### Data collection instrument and procedure

Data regarding on participants' socio-demographic, obstetric characteristics, alcohol consumption, awareness about the harmful effect of alcohol and the use of other substances (i.e.

Khat chewing and cigarette smoking) were collected using a structured questionnaire developed from previous literature.

However, family social support was assessed through the Multidimensional Scale of Perceived Social Support (MSPSSf). MSPSS has three subscales (family, friends and significant others), with a total of 12 items each scaled from 1 (very strongly disagree) to 7 (very strongly agree) [49, 50]. Respondents scored higher than 14 from a multidimensional scale of perceived family social support (MSPSSF) 7 Likert scale were recognized as having family social support, those who scored less than or equal to 14 were not having social support.

Likewise, the amount and frequency of alcohol consumption per day, week and month during the time of pregnancy were adapted from AUDIT and other articles [13, 21, 51]. In Ethiopian, both traditional (e.g. Tella, Araki, Teje) and manufactured (Beer, Wine) alcohol beverages are widely produced and consumed. People used local equipment's called Tella Birecheko, Melekia, and Birelie to drink traditional alcohols that can measure around 340 ml, 40 ml and 270 ml, respectively. Pregnant women were asked to estimate the average number of drinks they consumed. Finally, the data were standardized using the percentage and volume of alcohol in each ml of beverages. And drinking four more units of alcohol on one occasion was considered binge drinking.

Six questions were developed to assess the awareness of the harmful effect of alcohol on fetus, and those who answered the six questions correctly identified as having awareness about the consequences of alcohol. Furthermore, any amount of khat use and cigarette smoking even at once during the current pregnancy is known as pregnant women who chew khat and smoke cigarette, respectively. The questionnaire was prepared initially in English and translated to Amharic then back to English to check the consistency of the questions.

Eleven BSc nurses as a data collector and two public health officers as supervisor participated during data collection. Ethical clearance was obtained from Addis Ababa University, public health school ethical review committee. The right participants were systematically selected by data collectors when pregnant women came to maternal care. Data collectors explained the objective and procedures of the study, and their right to refuse or discontinue the interview at any time to participants. And, they knew that the questionnaire was anonymous and their privacy kept confidential. Also, Participants were aware of as the study has not any risk or direct benefit like incentives for them; it was only conducted for the achievement of objectives, and ultimately for improving health services of pregnant women and the community as a whole. After participants heard and understood all the necessary information and consent statement, they have been asked about their willingness to be interviewed, which was verbal consent. Those volunteers were again asked to write their name and put their signature with the date of interview in a prepared blank space found below the information sheet and statement of consent, which was written consent. For participants under 18 years old, informed consent was also obtained from their parents.

After that, data collectors interviewed the selected participant in the face to face manner by using a structured questionnaire at the waiting room of the maternal care clinic that took around 15 minutes. Those who found risky of alcohol consumption has been linked to substance abuse clinic, and responsible leaders in selected health centers and maternal care clinics were recommended and noticed to initiate and strengthen prevention of alcohol consumption among pregnant women.

## Data quality, processing and analysis

To assure the data quality training was given to the data collectors and supervisors for three days about the data collection tool, way of interview, and the ethical principles of

confidentiality before their involvement to the data collection. The collected data were checked for its completeness and consistency daily. Then, the collected data were checked, cleaned and entered into Epi. DATA 3.1 version and exported to Statistical Package for the Social Sciences (SPSS) version 20 for analysis. Descriptive statistics were summarized using frequencies, percentage, graphs, the mean and standard deviation. Binary logistic regression was employed to assess the association of each independent variables with the outcome variable. All variables with p-value < 0.2 during binary logistic regression were considered in the multiple logistic regression. The findings were presented using Odds ratio (crude and adjusted) with its 95% CIs. Moreover, a p-value < 0.05 was considered to declare the variable was statistically significant during the final multiple logistic regression analysis.

## Results and discussion

### Results

**Description of socio-demographic characteristics.** A total of 367 pregnant women were interviewed that made 92.44% response rate. This study revealed almost three fourth (74.7%) of participants were married. Seven out of ten (70.8%) participants were non-educated, whereas women who completed secondary education and above were 26 (7.1%). The age of participants ranged between 15 and 49 with mean age 27.43 (SD ± 4.77) years. Regarding ethnicity, Amhara and Oromo comprise the majority, 114 (31.1%) and 107(29.2%) respectively. Nearly half of the participants (48.2%) were followers of orthodox religion followed by Muslims (35.7%). The majority (42.5%) of pregnant women were housewives. Similarly, one-third of participants' monthly average income was between1001-2000 ETB (Table 1).

**Assessment of obstetric history of respondents.** As shown in Table 2 below, the majority of pregnant women (40.3%) were in the second trimester. Almost three quarters (73%) of pregnant women had a planned pregnancy. Three or more out of ten (31.6%) of pregnant women had a history of abortion.

Though most of the participants were in the second and third trimester, the proportion of alcohol consumption value was high among first trimester 21 (55.26%) (Fig 1).

**Alcohol consumption, binge drinking and other substances during pregnancy.** As illustrated in (Table 3) below, 146(39.8%) of pregnant women drank alcohol. The majority (44.4%) of participants predominately drank Tella. Fifty-four (14.71%) of participants consumed alcohol 2–4 times per month, while 15 (4.09%) of pregnant women drank four and above times per week. On the other hand, 13 (3.54%) of women had the experience of binge drinking during their pregnancy.

**Awareness about the harmful effect of alcohol use.** As demonstrated in Fig 2, Two hundred forty-four (66.5%) pregnant women have awareness, answered all six questions correctly regarding the harmful effect of alcohol consumption. Among those who have awareness, only 69 (28.28%) consumed alcohol. Whereas, of those who answered the awareness questions incorrectly, 76 (62.6%) drank alcohol.

**Factors associated with alcohol consumption among pregnant women.** As shown in Table 4, educational status, occupation, plan of pregnancy, history of abortion, awareness about the harmful effect of alcohol, and family social support of pregnant women were significant predictors of alcohol consumption among pregnant women. The higher odds alcohol consumption was observed among non-educated and attained primary education pregnant women with (AOR 95% CI = 8.47 (2.42, 29.62) and AOR 95% CI = 4.26 (1.23, 14.74), respectively compared to those completed secondary education and above.

Pregnant women who were housewives had a fourfold risk of alcohol consumption (AOR 95% CI = 4.18 (2.13, 8.22) compared to employed women. Women whose last pregnancy was

**Table 1. Socio-demographic characteristics of pregnant women in Kolfe Keranio sub city (n = 367), Addis Ababa, Ethiopia.**

| Variables | Frequency | Percent |
|---|---|---|
| **Age** | | |
| 15–19 | 8 | 2.2 |
| 20–24 | 100 | 27.2 |
| 25–29 | 117 | 31.9 |
| 30–34 | 106 | 28.9 |
| ≥35 | 36 | 9.8 |
| Mean (SD) | 27.43±4.775 | |
| **Marital status** | | |
| Married | 247 | 67.3 |
| Single | 81 | 22.1 |
| Divorced | 31 | 8.4 |
| Widowed | 8 | 2.2 |
| **Ethnicity** | | |
| Amhara | 114 | 31.1 |
| Oromo | 107 | 29.2 |
| Gurage | 91 | 24.8 |
| Tigrie | 32 | 8.7 |
| Others | 23 | 6.3 |
| **Religion** | | |
| Orthodox | 177 | 48.2 |
| Muslim | 131 | 35.7 |
| Protestant | 40 | 10.9 |
| Catholic | 10 | 2.7 |
| Others | 9 | 2.5 |
| **Educational status** | | |
| Not educated | 127 | 34.6 |
| Primary education | 123 | 33.5 |
| Secondary education | 91 | 24.8 |
| Above secondary education | 26 | 7.1 |
| **Occupation** | | |
| Employed | 105 | 28.6 |
| Housewife | 156 | 42.5 |
| Own business | 106 | 28.9 |
| **Average monthly house hold income** | | |
| ≤ 1000 Birr | 44 | 12 |
| 1001–2000 | 109 | 29.7 |
| 2001–3000 | 76 | 20.7 |
| 3001–4000 | 61 | 16.6 |
| >4000 | 77 | 21 |

unplanned had a two folds risk of alcohol consumption than their counters (AOR 95% CI = 2.47(1.33, 4.60). Likewise, pregnant women who had a history of abortion were prone to three-fold risk for alcohol consumption compared to pregnant women who hadn't [AOR 95% CI = 3.33 (1.33, 6.05)]. Women who had awareness about the effect of alcohol on fetus and them found to be a protective factor for alcohol consumption compared to those who had no awareness [AOR 95% CI 4.66 (2.53, 8.61)]. Moreover, in assessment family social support,

**Table 2. Obstetric history of the pregnant women in kolfe Keranio sub city (n = 367) Addis Ababa, Ethiopia.**

| Variables | Frequency | Percent |
| --- | --- | --- |
| **Trimester of pregnancy** | | |
| First trimester | 38 | 10.4 |
| Second trimester | 148 | 40.3 |
| Third trimester | 131 | 35.7 |
| **Number of pregnancy(gravidity)** | | |
| One | 88 | 24 |
| Two | 148 | 40.3 |
| ≥ Three | 131 | 35.7 |
| **Number of alive children** | | |
| No child | 112 | 30.5 |
| One | 134 | 36.5 |
| Two | 65 | 17.7 |
| > Two | 56 | 15.3 |
| **Planned pregnancy** | | |
| Yes | 268 | 73 |
| No | 99 | 27 |
| **History of abortion** | | |
| Yes | 116 | 31.6 |
| No | 251 | 68.4 |

more than half (66.5%) of them scored greater than or equal to 14 from the total score of MSPSSf. This study also found pregnant women who hadn't family social support had a two-fold risk for alcohol consumption [AOR 95% CI = 2 (1.14, 3.53) compared to those who had family social support.

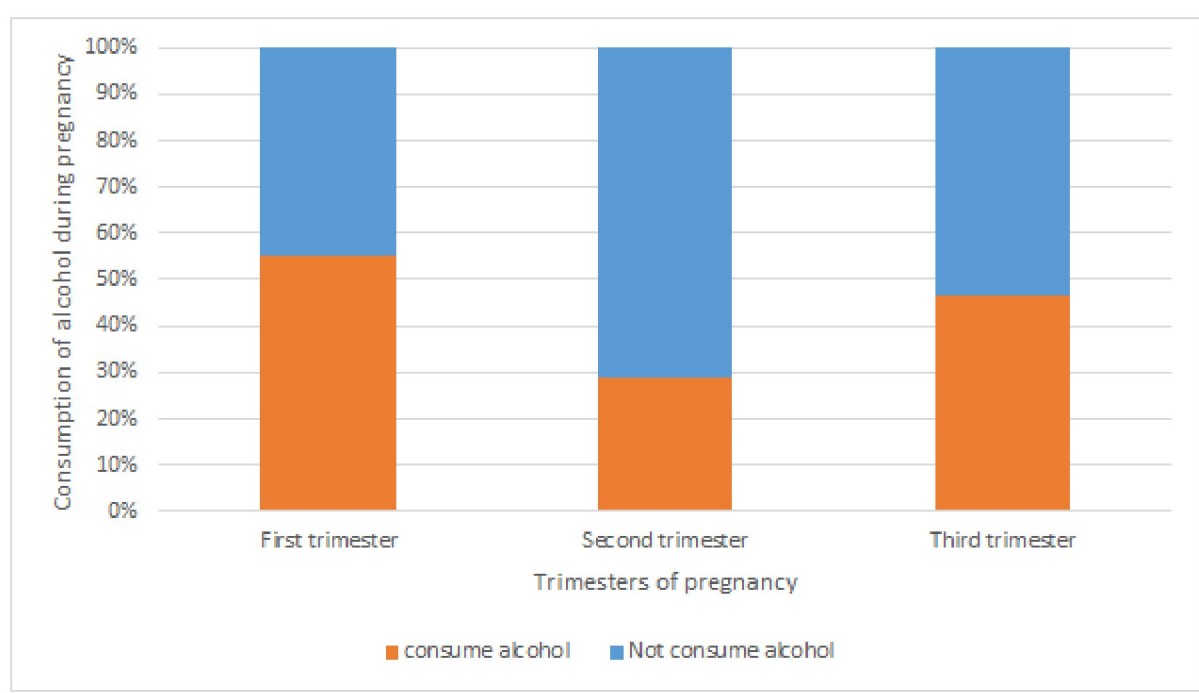

**Fig 1. Alcohol consumption among pregnant women respected to trimesters of pregnancy in Kolfe Keranio sub city (n = 367), Addis Ababa, Ethiopia.**

**Table 3. Alcohol consumption and binge drinking among pregnant women in kolfe sub city (n = 367), Addis Ababa, Ethiopia.**

| Variables | Frequency | Percent) |
|---|---|---|
| **Alcohol consumption before pregnancy** | | |
| Yes | 179 | 48.8 |
| No | 188 | 51.2 |
| **Alcohol consumption during pregnancy** | | |
| Yes | 146 | 39.8 |
| No | 221 | 60.2 |
| **Binge drinking during pregnancy** | | |
| Yes | 13 | 3.54 |
| No | 354 | 96.46 |
| **Frequency of alcohol consumption during pregnancy** | | |
| ≤ once a month | 36 | 9.81 |
| 2–4 times per month | 54 | 14.71 |
| 2-3times per week | 41 | 11.17 |
| ≥ 4 times per week | 15 | 4.09 |
| **Type of alcohol use** | | |
| Tella | 65 | 44.4 |
| Teje | 9 | 6.3 |
| Areki | 14 | 9.7 |
| Beer | 41 | 28.5 |
| Wine | 9 | 6.3 |
| Others | 7 | 4.9 |
| **Cigarette smoking during pregnancy** | | |
| Yes | 10 | 2.7 |
| No | 357 | 97.3 |
| **Khat use during pregnancy** | | |
| Yes | 39 | 10.6 |
| No | 328 | 89.4 |

## Discussion

The overall prevalence of alcohol consumption among pregnant women was 39.78% (34.74, 44.99). The finding from this study is lower than Ethiopia Demographic and Health Survey (45%) [52]. However, the prevalence of alcohol consumption reported in this study was higher than research conducted in Jimma hospital and Bahir Dar dwellers which are 31.3% and 34%, respectively [48, 53]. The possible reason could be the time of the study; population difference, and study design. A study conducted in Bahir Dar was community-based, unlike the current study which was institution-based.

This study's alcohol consumption prevalence was also higher than the findings from some of the African countries such as 16% in Uganda [54], 19.6% in South Africa [21], 25% in Uganda [22], 32.5% in Congo [23], 20.4% in Ghana [55] and 22.6% in South Nigeria [38].

Similarly, the magnitude from this study was higher compared to some studies in western countries like the United Kingdom (28.5%), Russia (26.5%) and Switzerland (20.9%) [19]. However, it is in line with a study conducted in Geneva that showed around 36.3% of the women drank at least one glass of alcohol during pregnancy [20]. In contrast to the above, alcohol consumption in this study was far lower compared to the results in Nigeria (59.3%) [26], Ghana (48%) [24], and Australia (56%) [25]. The observed discrepancy might be due to

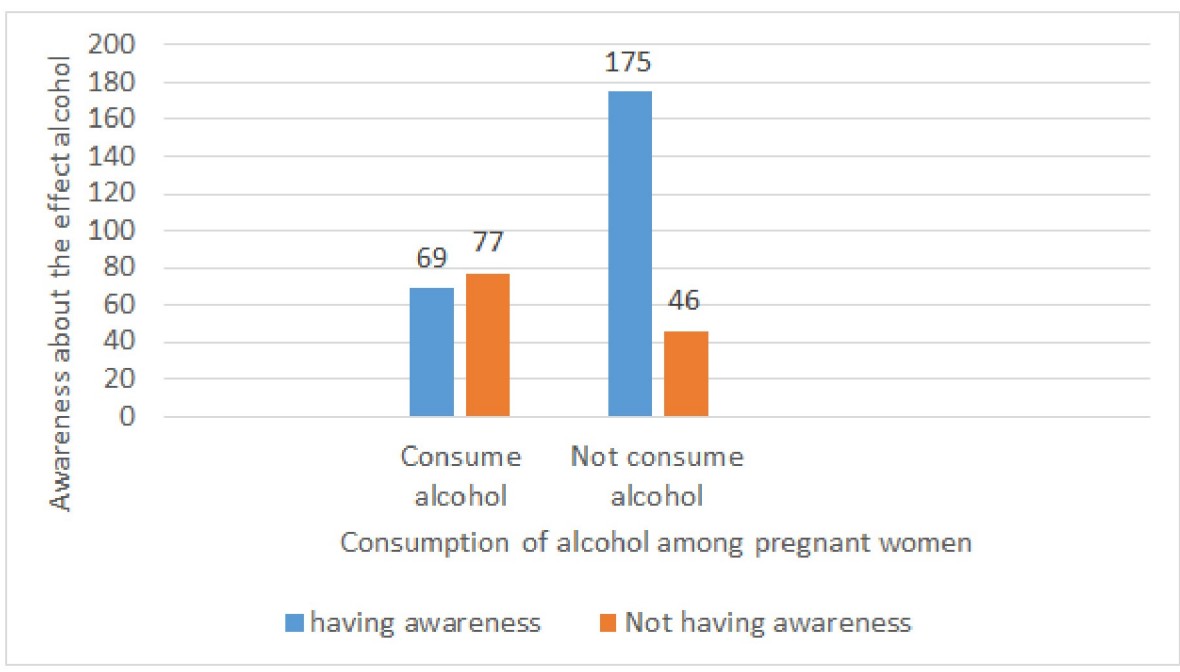

**Fig 2. Magnitude of alcohol consumption among pregnant women respected to awareness about the harmful effect of alcohol on fetus in Kolfe Keranio sub city (n = 367), Addis Ababa, Ethiopia.**

the type of alcohol consumed, study period, awareness and perception about the effect of alcohol on the fetus, and socio-cultural difference among the participants.

The magnitude of binge drinking, which is known for its hazardous effect on the fetus, was 3.54% (1.89, 5.98). It was similar to the findings to the studies in America (3.1%) [28], Canada (3%) [27], and South Africa (4.6%) [29]. However, it was much lower than the findings from studies in New Zealand (9%) [30], Uganda (10%), Congo (20.16%) [31], and Congo (25.42%) [23].

Another pattern of alcohol consumption, hazardous drinking which is found as a risk factor for abortion and miscarriage, was also determined in this study. It has still an immense negative health effect on the fetus. Of those pregnant women who consumed alcohol, 4.9% (2.03, 6.65) of them drank four or more unit of alcohol per week. Similarly, 11.7% of pregnant women in this study reported as they drank 2 to 3 unit of alcohol per week. Around 14% of participants were drinking 2 to 3 units of alcohol per month. A similar study conducted in New Zealand revealed 10% of pregnant women drank 7 units or more of alcohol within a week which was higher than the current study [30]. The discrepancy of the magnitude of binge drinking and amount of alcohol taken per week observed might be due to socio-demographic difference, study design, year of study, and awareness and perception about the effect of alcohol on the fetus.

This study also showed a significant association between not having formal education and alcohol consumption. It is in line with findings from previous studies [38, 53, 56]. It could be due to non-educated women might not have awareness on the harmful effect of alcohol consumption. However, another study showed that highly educated pregnant women were more likely to consume alcohol [32].

The occupation was also significantly associated with alcohol consumption. Being a housewife was shown as risky for alcohol consumption compared to being employed and working

**Table 4. Determinants of alcohol consumption among pregnant women in Kolfe Keranio sub city Addis Ababa, Ethiopia (n = 367).**

| Variables | Alcohol consumption during pregnancy | | | |
|---|---|---|---|---|
| | Yes | No | COR (95% CI) | A OR (95% CI) |
| **Religion** | | | | |
| Orthodox | 76 | 101 | 1.00 | 1.00 |
| Islam | 43 | 88 | 0.65 (0.41, 1.04) | 1.25 (042, 2.13) |
| Protestant | 24 | 16 | 1.99 (0.99, 4.01) | 3.01 (0.96, 5.02) |
| Catholic | 2 | 8 | 0.33 (0.69, 1.61) | 0.23 (0.11, 1.71) |
| Others | 1 | 8 | 0.166 (0.02, 1.36) | 7.09 (0.68, 18.4) |
| **Educational status** | | | | |
| Not educated | 65 | 62 | 4.40 (1.56, 12.40) | 8.47 (2.42, 29.62)* |
| Primary education | 61 | 62 | 4.13 (1.47, 11.66) | 4.26 (1.23, 14.74)* |
| Secondary education | 15 | 76 | 0.83 (0.27, 2.54) | 1.41 (0.38, 5.23) |
| Above secondary education | 5 | 21 | 1.00 | 1.00 |
| **Occupation** | | | | |
| Employed | 34 | 71 | 1.00 | 1.00 |
| housewife | 87 | 69 | 2.63 (1.57, 4.41) | 4.18 (2.13, 8.22)* |
| own business | 25 | 81 | 0.65 (0.35, 1.18) | 0.76 (0.35, 1.66) |
| **Plan of pregnancy** | | | | |
| Yes | 85 | 183 | 1.00 | 1.00 |
| No | 61 | 38 | 3.46 (2.14, 5.58) | 2.47 (1.33, 4.60)* |
| **Trimesters of pregnancy** | | | | |
| First | 21 | 17 | 1.41 (0.69, 2.85) | 1.01 (0.49, 5.11) |
| Second | 46 | 114 | 0.46 (0.29, 0.73) | 0.81 (0.12, 3.00) |
| Third | 79 | 90 | 1.00 | 1.00 |
| **Number of pregnancies** | | | | |
| One | 32 | 56 | 1.00 | 1.00 |
| Two | 56 | 92 | 1.06 (0.61, 1.84) | 1.77 (0.51, 7.42) |
| ≥ Three | 58 | 73 | 1.39 (0.79, 2.42) | 3.53 (0.85, 14.43) |
| **History of abortion** | | | | |
| Yes | 66 | 50 | 2.82 (1.79, 4.44) | 3.33 (1.33, 6.05)* |
| No | 80 | 171 | 1.00 | 1.00 |
| **Awareness about effect of alcohol** | | | | |
| Yes | 69 | 175 | 1.00 | 1.00 |
| No | 77 | 46 | 4.25 (2.68, 6.72) | 4.66 (2.53, 8.61)* |
| **Family social support** | | | | |
| Have no family social Support | 65 | 58 | 2.25 (1.45, 3.51) | 2 (1.14, 3.53)* |
| Have family social support | 81 | 163 | 1.00 | 1.00 |

Note.

* refers significant association.

their businesses. This finding is supported by other studies [48, 57]. It could be explained by housewives dependent behavior on their spouse that could create distress and their usual engagement in a routine home activity that includes preparing traditional alcohol beverages. Whereas, studies in European countries revealed that employed pregnant women are more likely prone to alcohol consumption [32]. Again, pregnant women lacking awareness about the harmful effect of alcohol on the fetus were more likely to consume higher levels of alcohol

than pregnant women who have the awareness. This finding is supported by other similar studies [36, 38, 55]. The other important factor significantly associated with alcohol consumption is family social support. According to the current study, pregnant women who had not family social support had higher odds of alcohol consumption which is consistent with other studies [33, 34]. But, a study conducted in Boston, Massachusetts revealed that social support is not a factor for alcohol consumption [58].

Moreover, a woman whose pregnancy was unplanned and had abortion history had a higher chance of consuming alcohol. This finding is also in line with the result of other studies [35–37, 48]. The possible reason could be that grief due to the previous abortion, psychological and physiological effect of abortion that might eventually cause distress and alcohol consumption.

Unlike some previous studies [35, 38, 59, 60], the current study didn't show a significant association between women age, gestational age, gravidity, ethnicity, number of alive children, religion, average monthly income, previous alcohol use, chat use, and cigarette smoking with alcohol consumption of pregnant women. Even though alcohol consumption is culturally accepted, the amount and frequency of alcohol drinking may be under-reported by pregnant women because of the possible perceived stigma and recall bias during the interview. Likewise, estimating the amount of alcohol drank by participants and changing the different type of alcohol into the standard unit was challenging and may not be accurate. It is also a real challenge and considered as a limitation for alcohol and other substances study across the world that includes the current research.

## Conclusions

This study revealed a higher magnitude of alcohol consumption among pregnant women. The prevalence of binge drinking and the amount and frequency of drinking per week among pregnant women was also higher. Factors such as not being educated, being housewife, having unplanned current pregnancy, having a history of abortion, not having awareness about the effect of alcohol on the fetus and not having family social support were significantly associated with alcohol consumption. Interventions focused on creating awareness about the harmful effects of alcohol on pregnancy; strengthening social support, and family planning services have to be developed and implemented. Moreover, we recommend to future researchers to conduct longitudinal study considering devices that help to measure the amount of alcohol in the blood.

## Acknowledgments

We would like to acknowledge Addis Ababa University, School of public health for their courage to begin this program and keep it to be continued. We also thank the staffs, and plan and program officers of Kolfe Keranio sub-city health centers for providing the necessary information, and their permission and support to conduct this research.

## Author Contributions

**Conceptualization:** Mezinew Sintayehu Bitew, Maereg Fekade Zewde.

**Data curation:** Maereg Fekade Zewde, Muluken Wubetu.

**Formal analysis:** Mezinew Sintayehu Bitew, Maereg Fekade Zewde, Muluken Wubetu, Addisu Alehegn Alemu.

**Funding acquisition:** Maereg Fekade Zewde.

**Investigation:** Maereg Fekade Zewde.

**Methodology:** Mezinew Sintayehu Bitew, Maereg Fekade Zewde, Muluken Wubetu, Addisu Alehegn Alemu.

**Project administration:** Mezinew Sintayehu Bitew, Maereg Fekade Zewde.

**Resources:** Maereg Fekade Zewde, Muluken Wubetu, Addisu Alehegn Alemu.

**Software:** Mezinew Sintayehu Bitew, Maereg Fekade Zewde, Muluken Wubetu, Addisu Alehegn Alemu.

**Supervision:** Mezinew Sintayehu Bitew, Maereg Fekade Zewde, Muluken Wubetu, Addisu Alehegn Alemu.

**Validation:** Maereg Fekade Zewde.

**Visualization:** Maereg Fekade Zewde.

**Writing – original draft:** Mezinew Sintayehu Bitew, Maereg Fekade Zewde.

**Writing – review & editing:** Mezinew Sintayehu Bitew.

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
