## [Decision Letter · Decision Letter 0]

19 Aug 2020

PONE-D-20-19931

Consumption of Alcohol and Binge Drinking among Pregnant Women in Health Institutions, Addis Ababa, Ethiopia: Prevalence and Determinant factors

PLOS ONE

Dear Dr. Bitew,

Thank you for submitting your manuscript to PLOS ONE. After careful consideration, we feel that it has merit but does not fully meet PLOS ONE’s publication criteria as it currently stands. Therefore, we invite you to submit a revised version of the manuscript that addresses the points raised during the review process.

We look forward to receiving your revised manuscript.

Kind regards,

Yukiko Washio, Ph.D.

Academic Editor

PLOS ONE

Journal Requirements:

2. During our internal evaluation we noted that participants provided verbal consent, did the ethics committees/IRBs approve this consent procedure?

Please explain: i) Why was written consent not obtained?  ii) How did you record/document participant consent?

*In the methods section it is stated that “Pregnant women, age greater than and equal to 18, in all trimesters attending the eleven kolfe Keranio sub-city health centers during the study period were included”. In the ethics statement it is stated that “written informed consent was also got from their parents for participants under 18 years old”. These statements appear to be contradictory. Please confirm whether your study included any participants under the age of 18 years and amend the manuscript as necessary.

3. In the abstract it is stated that the study “was conducted from April to June 2017”, but in the methods section of the manuscript it is stated that the study “was conducted from May to June 2017”. Please clarify this discrepancy.

Additional Editor Comments (if provided):

Reviewers' comments:

Reviewer's Responses to Questions

**Comments to the Author**

1. Is the manuscript technically sound, and do the data support the conclusions?

Reviewer #1: Yes

Reviewer #2: Yes

2. Has the statistical analysis been performed appropriately and rigorously? 

Reviewer #1: I Don't Know

Reviewer #2: Yes

3. Have the authors made all data underlying the findings in their manuscript fully available?

Reviewer #1: Yes

Reviewer #2: Yes

4. Is the manuscript presented in an intelligible fashion and written in standard English?

Reviewer #1: No

Reviewer #2: No

5. Review Comments to the Author

Reviewer #1: Comments to the Authors:

This research article has several strengths including addressing a significant public health concern with regard to alcohol use among pregnant women and identifying determinant factors. A key number of questions were examined in this study and the authors are commended for their important efforts. There are several recommendations for revision and these are highlighted below:

Introduction

1. Are there national survey data (prevalence rates) for alcohol/drug use in the general population that can be included in the Background section, to provide some context for why this is a public health concern in Ethiopia?

2. Binge drinking is defined twice in the Background (line 73, and line 92) – it is only needed once.

Methods

1. It is not clear how data was collected – were women approached while in the clinic? How was the study described to the women? Was there an incentive to complete study? How long did the interview take?

2. Were any participants currently in treatment for their alcohol use?

Results and Discussion

1. The response rate of 92% is impressive. Were there any missing data and if so, how was that handled?

2. It may strengthen the Limitations/Discussion section to include obstacles to disclosing alcohol use, such as stigma, bias, and concerns about potential threat of punishment or legal consequences that may exist for these women?

An additional note to the Authors: There are some sentences in the manuscript that are not clear, perhaps due to grammatical errors.

Reviewer #2: The manuscript submitted is an important contribution to the literature. I have noted some concerns in the text and have provided detailed comments there. The manuscript will need some extensive revision in sentence construction and language. At times it is very unclear and hard to follow. the presentation at times also seems more suitable for a thesis and has not been adequately adapted for a scientific article if indeed this data/paper is from a thesis (such as the aims and operational definitions). There are parts of the results which i also felt can be presented differently such as the Awareness about the harmful effect of alcohol use and Assessment of family social support of respondents. These are subheadings with only one sentence giving prevalence which could just be reported in table 4 - i have highlighted this in the text.

The paper would greatly benefit from a thorough review and more concise presentation.

6. PLOS authors have the option to publish the peer review history of their article (what does this mean?). If published, this will include your full peer review and any attached files.

Reviewer #1: No

Reviewer #2: No

---

## [Author Response · Author response to Decision Letter 0]

11 Sep 2020

Thank you so much all of you, editors and the two reviewers, for reviewing the manuscript critically for the sake of publishing quality scientific research. 

Response to academic editor’s comment and questions

1. Please ensure that your manuscript meets PLOS ONE's style requirements, including those for file naming. The PLOS ONE style templates can be found

Answer: I made the manuscript in line with the requirements of PlOS ONE 

2. For a question -During our internal evaluation we noted that participants provided verbal consent, did the ethics committees/IRBs approve this consent procedure?

Answer: in the actual procedure of the work, a copy of the master thesis, mother document of this manuscript, it states that both verbal and written informed consent was obtained. Data collectors explained to participants about the objective of the study, procedures, and their right to refuse or discontinue the interview at any time and other important information. After participants heard and understood all the necessary information and consent statement, they have been asked about their willingness to be interviewed, which is verbal consent. Those volunteers were again asked to write their name and put their signature with the date of interview at a prepared blank space found below the information sheet and statement of consent, which is written consent. That means they provide both verbal and written informed consent. So, in the manuscript that states “only verbal consent” was an error and misunderstanding when it was written in research ethics subheading. Now, I accepted your comments and took a correction in this revised manuscript.

3. Contradictory of the message regarding pregnant women under age 18 in the section of exclusion criteria and ethical consideration.

Answer: Yes, pregnant women start from 15 years old were enrolled in the study. Therefore, I removed the message that excludes women under 18 years old from the section of exclusion criteria.

4. In the abstract it is stated that the study “was conducted from April to June 2017”, but in the methods section of the manuscript it is stated that the study “was conducted from May to June 2017”.

Answer: the right period of data collection was from May to June 2017. I amended it.

5. Requested to update ORCID ID.

Answer: I did it.

 Response for Reviewers

Reviewer 1

1. Are there national survey data (prevalence rates) for alcohol/drug use in the general population that can be included in the Background section, to provide some context for why this is a public health concern in Ethiopia?

Answer: There are only local single articles in a different part of the country, and all reports that it the public health problems like the other countries across the world. But, there is no national survey study. 

2. Strong comments in method section regarding how data collected, how women approached, how the research described, and about the destination of pregnant women who used alcohol in a risky way

Answer: I accept all the comments and incorporate the raised concerns. Data collectors explained the objective and procedures of the study, and their right to refuse or discontinue the interview at any time to participants. Participants were aware of as the study has not any risk or direct benefit like incentives for them; it was only conducted for the achievement of objectives, and ultimately for improving health services of pregnant women and the community as a whole. After that, data collectors interviewed the selected participant in the face to face manner by using a structured questionnaire at the waiting room of the maternal care clinic that took around 15 minutes. Those who found risky of alcohol consumption has been linked to substance abuse clinic, and responsible leaders in selected health centers and maternal care clinics were recommended and noticed to initiate and strengthen prevention of alcohol consumption among pregnant women.

3. A question regarding missing data?

Answer: From the beginning, we have been adding 10% of possible non-response participants on the calculated samples. The non-responses in a manuscript were refusals to participate but not missing data’s/values. Fortunately, the actual non-response rate was below the prior plan or expected non-response rate that couldn’t affect the analysis and conclusion.

4. A comment of adding stigma or any other thing that could hinder the disclosure of alcohol use.

Answer: I understand your concern, and I try to add some possible perceived stigma. But, as I mentioned in the background of the study, there is a cultural acceptance of alcohol use in all demographic population. Since alcohol is historical and frequently used in our culture, which may be associated with Orthodox Christianity religion, stigma and punishment are not found. 

Reviewer 2

Answer: I saw all questions, comment and suggestion on each page of the document or manuscript that you gave me. I thank you for all you did on a manuscript. And I found it is essential for me and scientific research as well. I accepted all the comments and suggested solutions that you wrote and marked, and I corrected it in a document accordingly. I also received a note that you told me to incorporate the independent sub-heading variable (social support) with the next subheading in table four (associated factors- subheadings). But, I didn’t do for the variable of awareness about the harmful effect of alcohol consumption because this has a figure that describes and presents a descriptive message different from table four (associated factors). And I minimized the method sub-headings and merged it.

---

## [Decision Letter · Decision Letter 1]

14 Oct 2020

PONE-D-20-19931R1

Consumption of Alcohol and Binge Drinking among Pregnant Women in Health Institutions, Addis Ababa, Ethiopia: Prevalence and Determinant factors

PLOS ONE

Dear Dr. Bitew,

Thank you for submitting your manuscript to PLOS ONE. After careful consideration, we feel that it has merit but does not fully meet PLOS ONE’s publication criteria as it currently stands. Therefore, we invite you to submit a revised version of the manuscript that addresses the points raised during the review process.

We look forward to receiving your revised manuscript.

Kind regards,

Yukiko Washio, Ph.D.

Academic Editor

PLOS ONE

Reviewers' comments:

Reviewer's Responses to Questions

**Comments to the Author**

1. If the authors have adequately addressed your comments raised in a previous round of review and you feel that this manuscript is now acceptable for publication, you may indicate that here to bypass the “Comments to the Author” section, enter your conflict of interest statement in the “Confidential to Editor” section, and submit your "Accept" recommendation.

Reviewer #1: All comments have been addressed

Reviewer #2: (No Response)

2. Is the manuscript technically sound, and do the data support the conclusions?

Reviewer #1: Yes

Reviewer #2: Partly

3. Has the statistical analysis been performed appropriately and rigorously? 

Reviewer #1: I Don't Know

Reviewer #2: Yes

4. Have the authors made all data underlying the findings in their manuscript fully available?

Reviewer #1: Yes

Reviewer #2: No

5. Is the manuscript presented in an intelligible fashion and written in standard English?

Reviewer #1: No

Reviewer #2: No

6. Review Comments to the Author

Reviewer #1: The authors made helpful revisions to the manuscript based on reviewer's comments, thank you for your efforts.

Reviewer #2: While most of my comments have been addressed there are still some comments that i have attached that need to be addressed.

7. PLOS authors have the option to publish the peer review history of their article (what does this mean?). If published, this will include your full peer review and any attached files.

Reviewer #1: No

Reviewer #2: No

---

## [Author Response · Author response to Decision Letter 1]

1 Nov 2020

Response to reviewers 

First, we thank you for the editors and reviewers!

For both reviewer 

1. We modified all the texts of a document and tried to present intelligently and write in a standard English 

For reviewer 2

We corrected all the comments and suggestions line by line you provide

1. Line 30-typing error

2. Line 52 and 53- grammar error

3. Line 64-verb problem

4. Line 69-symbol problem

5. Line 87 to 93- comments to remove and correct

6. 102 to104- ambiguous sentence

7. Line 107 and 108-

8. Line 113

9. Line 222

10. Line 228 and 229- it has not a problem; it is the title of figure 1 which should be placed below the paragraph of that message according to the guideline of PLoS one

11. Line 272 and 273- commented to remove

Thank you again!

---

## [Decision Letter · Decision Letter 2]

12 Nov 2020

PONE-D-20-19931R2

Consumption of alcohol and binge drinking among pregnant women in Addis Ababa, Ethiopia: prevalence and determinant factors

PLOS ONE

Dear Dr. Bitew,

Thank you for submitting your manuscript to PLOS ONE. After careful consideration, we feel that it has merit but does not fully meet PLOS ONE’s publication criteria as it currently stands. Therefore, we invite you to submit a revised version of the manuscript that addresses the points raised during the review process.

We look forward to receiving your revised manuscript.

Kind regards,

Yukiko Washio, Ph.D.

Academic Editor

PLOS ONE

Reviewers' comments:

Reviewer's Responses to Questions

**Comments to the Author**

1. If the authors have adequately addressed your comments raised in a previous round of review and you feel that this manuscript is now acceptable for publication, you may indicate that here to bypass the “Comments to the Author” section, enter your conflict of interest statement in the “Confidential to Editor” section, and submit your "Accept" recommendation.

Reviewer #1: All comments have been addressed

Reviewer #2: (No Response)

2. Is the manuscript technically sound, and do the data support the conclusions?

Reviewer #1: Yes

Reviewer #2: Yes

3. Has the statistical analysis been performed appropriately and rigorously? 

Reviewer #1: I Don't Know

Reviewer #2: I Don't Know

4. Have the authors made all data underlying the findings in their manuscript fully available?

Reviewer #1: Yes

Reviewer #2: Yes

5. Is the manuscript presented in an intelligible fashion and written in standard English?

Reviewer #1: Yes

Reviewer #2: No

6. Review Comments to the Author

Reviewer #1: The authors have made revisions to the manuscript and it has been strengthened by these improvements.

Reviewer #2: (No Response)

7. PLOS authors have the option to publish the peer review history of their article (what does this mean?). If published, this will include your full peer review and any attached files.

Reviewer #1: No

Reviewer #2: No

---

## [Author Response · Author response to Decision Letter 2]

15 Nov 2020

First, we thank you for the editors and reviewers!

For both reviewer 

1. We modified all the texts of a document, presented it intelligently and wrote in standard English. 

2. We performed statistical analysis appropriately and rigorously.

For reviewer 2

We modified all the texts of a document, presented it intelligently and wrote in standard English depending on your consecutive and valuable strong comments and suggestions you, the reviewers, provide. 

We corrected all the comments and suggestion you provide line by line in the following way.

1. Line 84- we corrected a sentence as you suggest 

2. Line 96- we remove an ambiguous and less important adjective word “Terrific”.

3. Line 206-we modified the phrase three-fourth as you comment. 

We hope this the time to accept the paper for publication!

Thank you all again!

---

## [Decision Letter · Decision Letter 3]

26 Nov 2020

Consumption of alcohol and binge drinking among pregnant women in Addis Ababa, Ethiopia: prevalence and determinant factors

PONE-D-20-19931R3

Dear Dr. Bitew,

We’re pleased to inform you that your manuscript has been judged scientifically suitable for publication and will be formally accepted for publication once it meets all outstanding technical requirements.

Kind regards,

Yukiko Washio, Ph.D.

Academic Editor

PLOS ONE

Additional Editor Comments (optional):

Reviewers' comments:

Reviewer's Responses to Questions

**Comments to the Author**

1. If the authors have adequately addressed your comments raised in a previous round of review and you feel that this manuscript is now acceptable for publication, you may indicate that here to bypass the “Comments to the Author” section, enter your conflict of interest statement in the “Confidential to Editor” section, and submit your "Accept" recommendation.

Reviewer #1: All comments have been addressed

Reviewer #2: All comments have been addressed

2. Is the manuscript technically sound, and do the data support the conclusions?

Reviewer #1: Yes

Reviewer #2: Yes

3. Has the statistical analysis been performed appropriately and rigorously? 

Reviewer #1: I Don't Know

Reviewer #2: I Don't Know

4. Have the authors made all data underlying the findings in their manuscript fully available?

Reviewer #1: Yes

Reviewer #2: Yes

5. Is the manuscript presented in an intelligible fashion and written in standard English?

Reviewer #1: Yes

Reviewer #2: Yes

6. Review Comments to the Author

Reviewer #1: The authors addressed concerns and the manuscript has been strengthened by these revisions.

Reviewer #2: (No Response)

7. PLOS authors have the option to publish the peer review history of their article (what does this mean?). If published, this will include your full peer review and any attached files.

Reviewer #1: No

Reviewer #2: No

---

## [Editor Report · Acceptance letter]

10 Dec 2020

PONE-D-20-19931R3 

Consumption of alcohol and binge drinking among pregnant women in Addis Ababa, Ethiopia: prevalence and determinant factors 

Dear Dr. Bitew:

I'm pleased to inform you that your manuscript has been deemed suitable for publication in PLOS ONE. Congratulations! Your manuscript is now with our production department. 

Kind regards, 

on behalf of

Dr. Yukiko Washio 

Academic Editor

PLOS ONE